# Effect of Plant Defenses and Plant Nutrients on the Performance of Specialist and Generalist Herbivores of *Datura*: A Macroevolutionary Study

**DOI:** 10.3390/plants12142611

**Published:** 2023-07-11

**Authors:** Eunice Kariñho-Betancourt, Alejandra Vázquez-Lobo, Juan Núñez-Farfán

**Affiliations:** 1Laboratorio de Genética Ecológica y Evolución, Instituto de Ecología, Universidad Nacional Autónoma de México, Circuito Exterior, Ciudad Universitaria, Mexico City 04510, Mexico; karinho.betancourt@gmail.com; 2Centro de Investigación en Biodiversidad y Conservación, Universidad Autónoma del Estado de Morelos, Cuernavaca 62209, Mexico; alejandra.vazquez@uaem.mx

**Keywords:** alkaloids, herbivore performance, coevolution, specialist herbivore, generalist herbivore, *Datura*, plant nutrients, *Lema daturaphila*

## Abstract

Macroevolutionary patterns in the association between plant species and their herbivores result from ecological divergence promoted by, among other factors, plants’ defenses and nutritional quality, and herbivore adaptations. Here, we assessed the performance of the herbivores *Lema trilineata daturaphila*, a trophic specialist on *Datura*, and *Spodoptera frugiperda*, a polyphagous pest herbivore, when fed with species of *Datura*. We used comparative phylogenetics and multivariate methods to examine the effects of *Datura* species’ tropane alkaloids, leaf trichomes, and plant macronutrients on the two herbivores´ performances (amount of food consumed, number of damaged leaves, larval biomass increment, and larval growth efficiency). The results indicate that species of *Datura* do vary in their general suitability as food host for the two herbivores. Overall, the specialist performs better than the generalist herbivore across *Datura* species, and performance of both herbivores is associated with suites of plant defenses and nutrient characteristics. Leaf trichomes and major alkaloids of the *Datura* species are strongly related to herbivores’ food consumption and biomass increase. Although hyoscyamine better predicts the key components of the performance of the specialist herbivore, scopolamine better predicts the performance of the generalist; however, only leaf trichomes are implicated in most performance components of the two herbivores. Nutrient quality more widely predicts the performance of the generalist herbivore. The contrasting effects of plant traits and the performances of herbivores could be related to adaptive differences to cope with plant toxins and achieve nutrient balance and evolutionary trade-offs and synergisms between plant traits to deal with a diverse community of herbivores.

## 1. Introduction

Coevolutionary theory predicts the contrasting performances of herbivores according to their evolutionary history in interaction with their host plants [1]. Trophic specialist herbivores, due to their historical association with the host plant, are expected to perform better with their host plant and other phylogenetically related species, owing to their ability to cope with the plant defenses [2]. In contrast, trophic generalist herbivores may lack fine-tuning adaptive mechanisms to cope with plant defenses of a non-host plant and be, in consequence, susceptible to their negative effects. However, some generalist herbivores possess the capability to feed on a diverse array of host plant species. Numerous studies have supported these predictions by assessing the diverse physiological countermeasures evolved by specialist herbivores to avoid, excrete, or metabolize host plant´s plant defenses [3,4]. In some instances, herbivores are capable of using plant chemical defenses as cues to locate their host plant or to gain protection against predators [5]. On the other hand, deterrent and/or toxic effects of plant secondary compounds on generalist herbivores have been reported [6]. Nevertheless, some generalist insects have also evolved the ability to consume highly toxic host plants [7,8], but specialist herbivores remain susceptible to and negatively affected by defensive compounds, suggesting ongoing arms races [9,10]. The link between plant defense traits and herbivore performance [11,12,13], as well as insects’ adaptation to such traits, has been experimentally demonstrated [10] but may be constrained by synergistic effects when defense is composed of multiple plant traits. Although evidence of synergisms and trade-offs between physical and/or chemical defenses has been documented whether combinations of defense traits are adaptive seems to be context dependent [14] Additionally, whether a single or a suite of defense plant traits (physical or chemical) act more effectively against specialist or generalist herbivores remains unclear [15]. The understanding of the mechanisms and processes underlying the evolution of plant–herbivore interaction and the evolution of diet breadth in insects makes necessary to assess the response of generalist and specialist herbivores to multiple plant traits on specific plant lineages. 

Besides plant defenses, plant nutritional quality to herbivores may determine the herbivore–host plant association. However, relatively few studies have assessed the interplay between plant defense traits and nutrients on herbivores’ performances [16,17], and even fewer have compared their effect on specialized and generalist herbivores across a plant phylogeny (see [18,19,20]). In this study, we examined the role of multiple plant traits on the performances of specialist and generalist herbivores using plant species from the genus *Datura*.

Species of *Datura* are known to produce a diverse array of highly toxic tropane alkaloids [21,22] that negatively affect herbivores [23,24] by inhibiting enzymatic activity and competing for muscarinic acetylcholine receptors [25,26]. Tropane alkaloids function as resistance traits against generalist and specialist herbivores within species by reducing leaf damage [27,28,29,30] and plant infestation [31]. At macroevolutionary scale, defense plant traits of *Datura* species, leaf trichomes, and major tropane alkaloids are correlated, suggesting an adaptive role for covarying traits [32]. However, the independent and/or joint effect of plant defenses and macronutrients on the performances of specialist and generalist herbivores has not been assessed using species of *Datura*. Here, using comparative phylogenetics, analytical chemistry, and large-scale experiments, we examine the effect of plant defenses and nutrients in *Datura* species on generalist and specialist insect herbivores’ performances. Specifically, we aimed to compare the performance of the main trophic specialist herbivore of *Datura, Lema trilineata daturaphila,* with that of *Spodoptera frugiperda*, a polyphagous pest herbivore, and to assess whether the herbivores’ performances are predicted by a single plant nutritional/defensive characteristic or by a suite of these (sensu [33]). We employed a large data survey of plants that included tropane alkaloids, leaf trichomes, and plant macronutrients, as well as the amount of food consumed (FC), number of damaged leaves (DL), larval biomass increment (BI), and larval growth efficiency (GE) to estimate herbivore performance. We assayed plants of *Datura* species derived from the same natural populations and grown under the same conditions as those from which the chemical and physical defenses were previously characterized by Kariñho-Betancourt et al. [32]. Thus, we combine our newly collected data of plant nutrients and herbivore performance with a set of previously collected measures of plant defensive traits to test: (1) whether herbivores’ performances are shaped by their life history, (2) if such performance is predicted by a single or a suite of defensive (chemical and/or physical traits) and nutritive traits, and (3) how these traits differ between the generalist and specialist insects.

## 2. Results

We assessed the variation in performance between *Lema trilineata daturaphila* and *Spodoptera frugiperda* when fed with different species of *Datura* by means of a mixed model ANOVA. The analysis indicates significant differences between the herbivores, *Datura* species, and the interaction of the type of herbivore x plant species (i.e., different response between herbivores to a given *Datura* species) (Table 1) for all performance components except for growth efficiency (see Methods). Although the generalist herbivore damaged more leaves (DL), the specialist herbivore outperformed the generalist in the rest of traits (Figure 1). Exceptions to this trend were *D. quercifolia* and *D. lanosa,* where the generalist herbivore outperformed the specialist in specific performance components including food consumption (FC) (Appendix A). The results also indicate that the species of *Datura* differ in their general effect on herbivore performance (Table 1, Appendix A). Moreover, *L. trilineata daturaphila* performs better than *S. frigiperda* across *Datura* species, except on *D. srtramonium, D. quercifolia*, and *D. inoxia*, where no difference was detected; however, both herbivores had poor performance on *D. inoxia* and *D. ferox* (see BI in Appendix A).

### Herbivore Performance and Plant Traits

To evaluate the relationship between plant traits and patterns of herbivore performance across *Datura* species, we performed a phylogenetic principal component analysis (pPCA) and found that pPC1 and pPC2 accounted for 85% of the variation (Table 2). In the first pPC, the greater positive loads (correlation coefficient > 0.5) corresponded to plant defense traits including scopolamine, atropine, and total alkaloid concentrations and nutrient-related traits such as nitrogen (N) and water content (Table 2). The greater negative loads (<0.5) corresponded to herbivore traits. The pPC1 separates *D. discolor, D. kymatocarpa, D. inoxia*, and *D. metel* from the other species. Eigenvectors of plant defensive traits tend to have similar orientations.

Eigenvectors of nutrients were displayed with different orientations over pPC2 (Figure 2). The pPC2 mainly separates *D. rebura, D. pruinosa*, and *D. wrightii*, in association with the performance components of the specialist herbivore *L. daturaphila* from *D. lanosa, D. stramonium, D. ferox*, and *D. quercifolia* in association with the components of the generalist herbivore *S. frugiperda* (Figure 2). Likewise, the variables for each herbivore showed a contrasting relationship to specific plant traits.

To examine the relationship between plant and herbivore variables, we performed phylogenetic correlations least squares (PGLS) analysis among plant traits and herbivore performance. Overall, the performance components of the specialist herbivore were positively associated with hyoscyamine and nitrogen (N) and negatively associated with scopolamine, atropine, total alkaloid concentration, and phosphorous (P). By contrast, the performance of the generalist herbivore *S. frugiperda* was positively associated with phosphorus (P) and negatively associated with hyoscyamine and water content (Figure 2). The analysis also shows contrasting associations for specific plant defenses and nutrients between the specialist and generalist herbivores. Scopolamine and phosphorus (P) were negatively correlated with the specialist performance, affecting the biomass increase and the number of damaged leaves. These plant traits showed the opposite relationship in the performance of the generalist herbivore. Carbon (C), nitrogen (N), and water content showed a negative correlation with the biomass increase, growth efficiency, and number of damaged leaves in the generalist herbivore. Unlike the chemical traits, leaf trichomes were only significantly (negatively) correlated with the performance of *S. frugiperda*.

In addition, we created models of all possible combinations of variables and calculated the value of Akaike’s information criterion (AIC) to identify the variable or set of variables that better predict/explain the herbivores’ performances. The permutation method indicated distinctive sets of traits that predicted specific performance components for each herbivore. The best fitted models of performance for the specialist herbivore included leaf trichomes, hyoscyamine, and scopolamine. However, the only significant model for explaining the variation in the number of damaged leaves by the specialist herbivore included scopolamine and leaf trichomes. For the generalist herbivore, the only significant model to explain the variance in the number of damaged leaves included total alkaloid content and leaf trichomes.

The models related to leaf nutrients indicate that food consumption by the specialist herbivore is better explained by carbon (C) and phosphorus (P). On the other hand, a combination of nitrogen (N), carbon (C), and water better explained most of the performance components of the generalist herbivore (Table 3).

## 3. Discussion

This study examined the performance of specialist and generalist herbivores when feeding on plant species of the alkaloid-producing genus *Datura.* We explored which defense- and nutrient-related traits, or *suites* of traits, better predicted the herbivores´ performances. In an expected overall result given their history of interaction, the specialist beetle *L. trilineata daturaphila* performed better than the generalist moth *S. frugiperda*. Both species displayed differential responses in terms of consumed leaf tissue and rate of weight increase depending on the *Datura* species. Even when the specialist herbivore outperformed the generalist in most performance components, the generalist consumed more leaves than the specialist and had a comparable biomass increase when consuming some *Datura* plant species. Although these differences in performance point to adaptive responses to plant defense, given that herbivores differ in many life history traits and phylogenetic positions, we cannot exclude that factors other than their diet range may account for the differences between the two herbivores. Nonetheless, within *Datura* species, both insects showed variable patterns of association with specific plant traits, suggesting a complex relationship between plant defense and nutrients to cope with herbivory.

### Herbivore Performance and Plant Traits

We found two patterns between herbivores: (1) the performance of both the specialist insect *L. trilineata daturaphila* and the generalist insect *S. frugiperda* is better predicted by a combination of contrasting pairs of defensive traits, including leaf trichomes and major alkaloids scopolamine and hyoscyamine, and (2) the performance of the generalist insect is explained by a wider set of nutrient-related traits, including the nitrogen (N), carbon (C), and water contents.

The results show that the best models to predict herbivore performance include the variable leaf trichomes for both types of herbivores (cf. Appendix A). A combination of hyoscyamine and trichomes better predicted the performance components of the specialist herbivore, including growth efficiency (GE) and food consumption (FC). However, biomass increment (BI), a key component of herbivore growth and performance, is predicted by scopolamine and trichomes in *L.t. daturaphila*, which also predicted most performance components of *S. frugiperda*, including number of damaged leaves (DL), which is usually related to the estimation of general plant resistance to herbivores (cf. Appendix A). Consistently across the genus *Datura*, we found specific suites of covarying defensive and nutrient-related traits antagonistically associated with the performance of the specialist and generalist herbivores. For instance, scopolamine and phosphorus (P) had a negative effect on food consumption and biomass increase in the specialized beetle, suggesting a synergistic effect to cope with a specific guild of herbivores. In contrast, hyoscyamine and leaf trichomes were positively associated with the performance of the specialist herbivore and leaf trichomes had a negative effect on the performance of the generalist moth, suggesting an antagonistic relationship with scopolamine. Overall, this evidence supports the synergy hypothesis that posits that some traits are more effective when co-expressed with others compared [33].

Recent studies have identified numerous synergies and antagonisms or syndromes among plant defenses and nutrients [34,35]. Such syndromes suggest that plants may face a compromise (trade-off) between defending themselves against generalist herbivores and withstanding the damage caused by coevolved specialists. Evidence of this has been documented across the phylogeny of *Oenothera* and *Gayophytum* [18]. Consistent with our results, in *Datura* a negative phylogenetic correlation (adaptive trade-off) between scopolamine and the syndrome of hyoscyamine and leaf trichomes has been documented [31]. Additionally, within *Datura* species, previous studies have documented that either single or suites of major alkaloids, including hyoscyamine (atropine) and scopolamine, and leaf trichomes play a key ecological role, functioning either as defense against both specialized and non-specialized herbivores and as cues for specialist herbivores (2327,28930). Thus, the evolutionary consequences of synergisms and trade-offs in plant defense are particularly important since the simultaneous evolution of different defensive traits can be constrained [36,37]. In this regard, the ecological and evolutionary outcomes of the *Datura*–herbivore interaction depend on the insect herbivore community structure, hence the selective pressures exerted on specific defenses.

The nutrition quality of plants substantially influences herbivore performance [17,38]. In our study, the performance of the generalist herbivore was highly affected by plant macronutrients. Although phosphorus (P) was positively associated with the generalist insect, affecting the number of consumed leaves, the other nutrients and water content showed a negative effect on most performance components for the generalist insect. Theoretically, although the diet heterogeneity of foraging insects may reduce growth and development rates, in some cases variation in plant food quality has no effect on the performance of polyphagous lepidopterans (e.g., [39]). Nonetheless, our results suggest that the generalist moth is particularly sensitive to carbon (C) intake, as this nutrient had greater weight in models and a strong negative effect on growth efficiency. Because the generalist herbivore fed on plants but lost weight afterwards most of the time, a mismatch between the food quality of plants (e.g., absence of specific essential nutrients) and the nutritional requirements of the generalist herbivore could explain the poor performance of the generalist insect. In addition to the effects of plant quality on polyphagous insects, empirical evidence indicates that a suboptimal nutritional balance may increase the effectiveness of plant defenses [40]. For instance, an optimal ratio of macronutrients seems to modulate the toxic effects of the alkaloid nicotine on the growth of the tobacco hornworm caterpillars *Manduca sexta* [41]. Hence, the pattern of consumption and efficiency of herbivores here documented may reflect the interaction between alkaloids, leaf trichomes, and macronutrients in the genus *Datura*.

## 4. Materials and Methods

### 4.1. Study System

The study system consisted of species of the genus *Datura* (Solanceae). In these plant species, we studied the pattern of herbivores’ food consumption and growth in relation to plant chemical and physical defenses and macronutrients. The genus *Datura* is composed of 13–14 species of annual herbs and perennial shrubs [42,43] that are native to the dry, temperate, and subtropical regions of America [44] and mainly distributed in Mexico [45]. In natural populations, the species of *Datura* are commonly consumed by both generalist and specialist herbivores, including the specialist folivore, the three-lined potato beetle *Lema trilineata daturaphila* (Coleoptera: Chrysomelidade) [30,46,47,48], and the generalist folivore, the fall armyworm *Spodoptera frugiperda* (Lepidoptera: Noctunidae) in Central Mexico (J. N-F pers. obs., 1987–1989).

*Datura* species exhibit great variation in leaf trichomes and tropane alkaloids within [29,30] and among species [32,49]. These traits have been implicated in plant resistance against natural enemies [27,28,50,51] and have shown differential effects based on herbivore’s life history. For example, whereas the tropane alkaloid scopolamine of *D. stramonium* reduces leaf consumption by the generalist herbivores *Helicoverpa zea* (corn earworm) and *Popilia japonica* (Japanese beetle), it increases consumption by the specialist flea beetle *Epitirix parvula* [23]. Nonetheless, the effect of *Datura* traits has not been tested on herbivores across phylogeny.

### 4.2. Experimental Design

Our study focuses on the performances of two leaf-chewing larvae with different life histories on *Datura*: (1) specialist herbivore—the three-lined potato beetle *L. trilineata daturaphila* (Coleoptera: Chrysomelidae) and (2) generalist herbivore—the fall armyworm, *Spodoptera frugiperda* (Lepidoptera: Noctunidae). Three-lined potato beetles are dietary specialists found on plants in the family Solanaceae, including species of the genus *Datura*. Adults are 7–8 mm long (1/4 inch) and orange yellow in color with three lengthwise black stripes. The females lay orange-yellow eggs in groups, usually on the undersides of leaves by veins. Larvae are slug-like with black heads (Figure 3), ranging between 1 and 6 mm in length. The larvae hatch in late June or July and tend to stay in groups as they feed on their host plant until pupation. During development, larvae can increase in mass by a total of 60 mg and consume an average total of 217.3 mg of food [52].

The polyphagous moth *S. frugiperda* is widely distributed in eastern and central North America and in South America; it forages in diverse habitats, mainly grasses and grain crops such as corn, but its diet might include plants belonging to 76 different plant families including Solanaceae [53]. Adult moths are 32–40 mm long wing tip to wing tip, with a brown or gray forewing and a white hindwing. Females prefer to lay eggs on the underside of leaves; the eggs usually hatch within a few days. The larvae pass through six different instars, each varying slightly in physical appearance and pattern [54]. The hatched larvae are lightly colored and become darker after the first instar (Figure 3). The larval stage lasts from 14 to 30 days, depending on temperature, and the insect reach more than 60 mm in length [55]. Larvae of this species are usually employed to test the effects of diverse chemical compounds and to develop crop management strategies such as in insecticide testing [56,57,58].

### 4.3. Rearing of Larvae

Eggs of *L. trilineata daturaphila* were collected from two natural populations of *D. stramonium* in central Mexico and reared on sterilized moist soil in sealed glass jars (5 L). Larvae were fed with fresh *D. stramonium* leaf tissue from naturally grown plants stored within the jar that were replaced daily until pupation (Appendix A). The larvae were grown at 25 ± 1 °C, >70% relative humidity, and with a 12:12 photoperiod of light:dark. After eclosion, the adults were transferred to different jars (under the same conditions) and allowed to mate during the next 15–20 d. The laid eggs (F1) were collected and transferred to a different jar. We followed the same protocol to obtain F2 larvae. *Spodoptera frugiperda* eggs from laboratory strains were settled in moist filter paper in sealed Petri dishes (9 cm). Hatched larvae were reared on an artificial diet [56,57] at 27 ± 1 °C, > 70% relative humidity, and with a 12:12 photoperiod of light:dark.

### 4.4. Bioassays

To perform the bioassays with both the specialist and generalist herbivore larvae, we employed 11 out of the 13 terrestrial species of *Datura*. The plants were grown in a greenhouse by sowing seeds from maternal families (natural progenies; [59]) into pots filled with sterilized commercial soil. The plants were watered daily and regularly rearranged at random to avoid positional effects. Since previous studies in *Datura* indicate that the phenology of most specialist and generalist herbivores is synchronized with plant reproduction [60,61], herbivores were exposed to experimental plants at the onset of flowering. We selected ten experimental plants of similar sizes per *Datura* species to feed larvae of each species of herbivore (i.e., specialist and generalist). Each plant was transferred to a single pot. For the experiment with the specialist herbivore, plants were spaced 60 cm away from each other to avoid contact among plants and thus the potential movement of larvae between plants.

*Lema trilineata daturaphila* larvae of the second to fourth instars were fed with leaf tissue from each *Datura* species. Larvae were starved for 12 h before feeding with each plant. Two larvae (F_2_) per plant (10 plants per species) were settled on the adaxial side of the leaves of each *Datura* species. Every larva was settled on a leaf located at opposite ends of the plant, and each leaf was marked with a distinctive color corresponding to each larva (larva 1: yellow; larva 2: blue) (Appendix A). A plastic stencil was overlaid on top of each pot, covering the soil, to prevent the larva burying into the soil. To accurately record the damage produced by the larvae, every leaf on which the larvae fed was color-marked and the feeding larvae were recorded and monitored every two hours. After 48 h, the larvae and leaves were collected.

Due to differences in life history, physiology, and behavior, the larvae of the specialist beetle and the generalist moth develop and relate differently to the host plant. Whereas *L.* larvae begin to feed immediately after being placed on the plant, the foraging larvae of *S. frugiperda* tend to explore for a long time before feeding and in many cases move away from the host plant. Therefore, to properly examine the effect of the host plant on the generalist herbivore, the protocol was modified and implemented as follows. *Spodoptera frugiperda* larvae were fed individually in a Petri dish. As in the experiment with the specialist herbivore, two larvae were fed with a single plant, for a total of ten plants per *Datura* species. Fresh leaf tissue from each *Datura* species was provided ad libitum and regularly replaced when withered during 48 h. At the end of the experiment, the larvae and leaves were collected. Although the differences in performance between the herbivores due to life history traits and metabolic characteristics are not discarded, this approach allowed us to evaluate the performances of both specialist and generalist herbivores in the same time frame while avoiding the effects of ontogenetic variation on plant traits; this also allowed us to preserve the adequate conditions for feeding and development for each herbivore. Since the differences in larval rearing could influence the estimates of the herbivores´ performances, we performed a pilot experiment and found that *S. frugiperda* larvae do not consume different amounts of food when either fed directly on the plant or within a Petri dish (*t*_1,58_ = 2.001, *p* = 0.512).

To measure the larval growth (g) of both the specialist and generalist herbivores, we weighed each larva at the start (0 h) and end (48 h) of the experiment. All larvae weights are expressed on a fresh weight basis. To estimate the total food/leaf tissue ingested (cm^2^) by each larva during the 48 h period, we measured the damaged area (A_D_) of all leaves consumed per larva by using a leaf area meter (Winfolia, Regent Instruments Inc., Quebec, QC, Canada).

### 4.5. Herbivore Measures

To identify the different components of larval performance, we split food consumption and utilization into three components [62,63].

Food consumption (cm^2^): FCRelative biomass increment: BI
$$\frac{\mathrm{final}\;\mathrm{weight}\;\left(\mathrm{wtf}\right)}{\mathrm{initial}\;\mathrm{weight}\;\left(\mathrm{wti}\right)}-1$$Growth efficiency (efficiency of the conversion of ingested food to body substance): GE
$$\frac{\mathrm{biomass}\;\mathrm{increment}\;\left(g\;\mathrm{wwt}\right)}{\mathrm{food}\;\mathrm{ingested}\;\left({\mathrm{cm}}^{2}\right)}$$In addition, as a measure of larva mobility during feeding and thus an indirect indicator of the amount of energy spent, we counted the number of leaves damaged by each larva: DL

### 4.6. Plant Measures

In order to quantify the leaf macronutrients in each *Datura* species, we prepared a sample of oven-dried ground leaves (6 g) for each species. These samples were used to analyze nitrogen (N), phosphorus (P), and carbon (C) using colorimetric techniques that included the molybdenum blue colorimetric method and malachite green assay preceded by the process of mineralization to obtain simple inorganic compounds [64]. Additionally, we recorded the fresh and dry weights of the leaves to calculate the water content (W) of each species.

To evaluate plant defensive traits, we obtained the average concentration for each of five leaf tropane alkaloids for each species (hyoscyamine, scopolamine, atropine, tropine, and solanine) and the average leaf trichome density for the leaves harvested at the reproductive stage for each species [32].

### 4.7. Statistical Analysis

To assess the variation in performance between the specialist and generalist herbivores and the effect of plant species on the components of herbivore performance, we performed a mixed model ANOVA of (1) food consumption, (2) the number of damaged leaves, (3) the relative biomass increment, and (4) growth efficiency as response variables. The plant species and type of herbivore (specialist and generalist) were the random and fixed effects, respectively.

To assess the relationship between plant traits and patterns of herbivore performance across *Datura* species, we used a molecular phylogeny of the 11 *Datura* species (Appendix A, Figure 4) constructed with chloroplast (*psb*A-trnH and *trn*L-*trn*F) and nuclear (ITS rRNA) datasets (see [32]). First, to examine the variance of plant traits in relation to herbivore performance, we performed a phylogenetic principal component analysis pPCA in the *R* statistical package using the *phytools* program. The phylogenetic correction in pPCA adjusts the covariance/correlation matrix to remove the expected phylogenetic correlation among traits. Hence, the axes in pPCA describe the non-phylogenetic component of shape covariance and, as a result, are independent from phylogeny and orthogonal to one another [65]. The pPCA included the plant defense and nutrient traits for *Datura* species and the herbivores’ (*L. daturaphila* and *S. frugiperda*) components of performance.

To identify a specific trait or set of traits that better explain the components of the herbivores’ performances, we executed a permutation procedure. We split the predictor variables into defense and nutrient traits. Based on the pPCA, only the most informative chemical traits (i.e., hyoscyamine, scopolamine, and total alkaloids) and leaf trichome density were included. We then correlated the plant traits with herbivore performance across the *Datura* species. Phylogenetic correlations were obtained using the generalized least squares (GLS) method for independent contrasts [66,67]. In addition, for each group of plant traits (nutrients/defenses) we created models of all possible combinations of variables (phylogenetic independent correlations) and calculated the value of Akaike’s information criterion (AIC) as a measure of model fit [68]. The model with the lowest AIC was the best out of all the models tested to explain herbivore performance.

## 5. Conclusions

This study provides the first experimental evidence across the *Datura* phylogeny on the effect of plant defense and nutrients on the performance of generalist and specialist insect herbivores. Variable patterns of association between the specialist herbivore *Lema trilineata daturaphila* or the generalist *Spodoptera frugiperda* and plant defenses and macronutrients showed a differential effect for plant defense and nutrients.

Our results suggest (1) adaptive differences between insect herbivores along the phylogeny of the host–plant genus, (2) evolutionary trade-offs and synergism between plant defenses and nutrients to cope with the diverse community of herbivores of *Datura*, and (3) an interplay of plant nutrients and defenses to reduce the nutritional balance of herbivores, thus increasing the effectiveness of defense. The patterns of herbivore performance here documented, reflect the evolutionary and ecological dynamics of the *Datura*–herbivore interaction and provide an integrative view of the relationships among different plant and herbivore traits.

## Figures and Tables

**Figure 1 plants-12-02611-f001:**
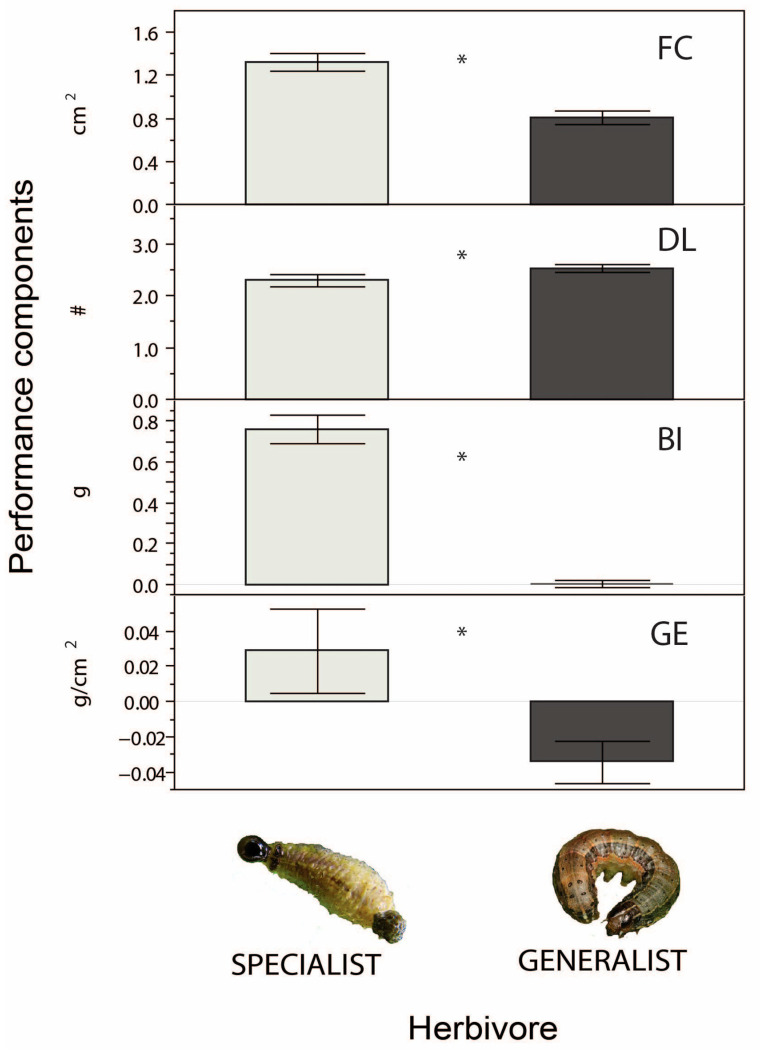
Variation in performance components of the specialist herbivore *Lema trilineata daturaphila* and the generalist herbivore *Spodoptera frugiperda*. Symbols: food consumption (FC), number of damaged leaves (DL) indicated by hash symbol, biomass increment (BI), and growth efficiency (GE), across eleven *Datura* spp. (*, *p* < 0.05).

**Figure 2 plants-12-02611-f002:**
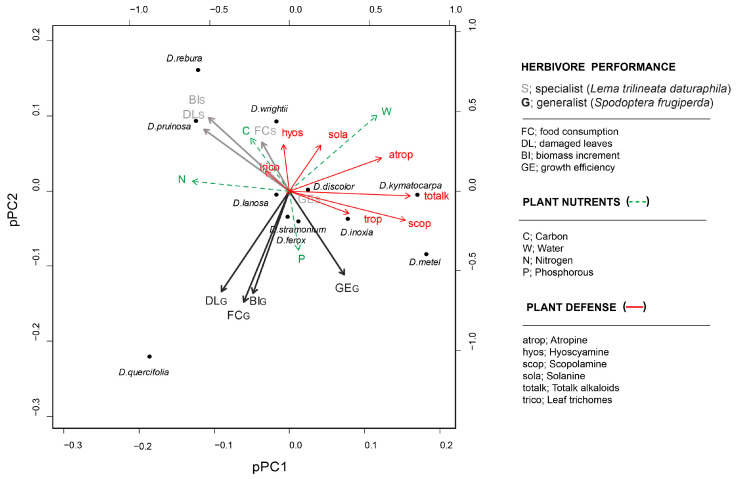
Phylogenetic principal components analysis (pPCA) of herbivore performance and plant nutrients and defenses.

**Figure 3 plants-12-02611-f003:**
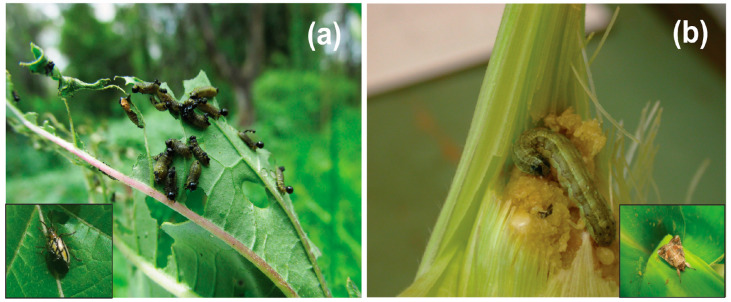
(**a**) Larvae of the specialist herbivore *Lema trilineata daturaphila* (Coleoptera: Chrysomelidae) and (**b**) the generalist herbivore *Spodoptera frugiperda* (Lepidoptera: Noctuidae).

**Figure 4 plants-12-02611-f004:**
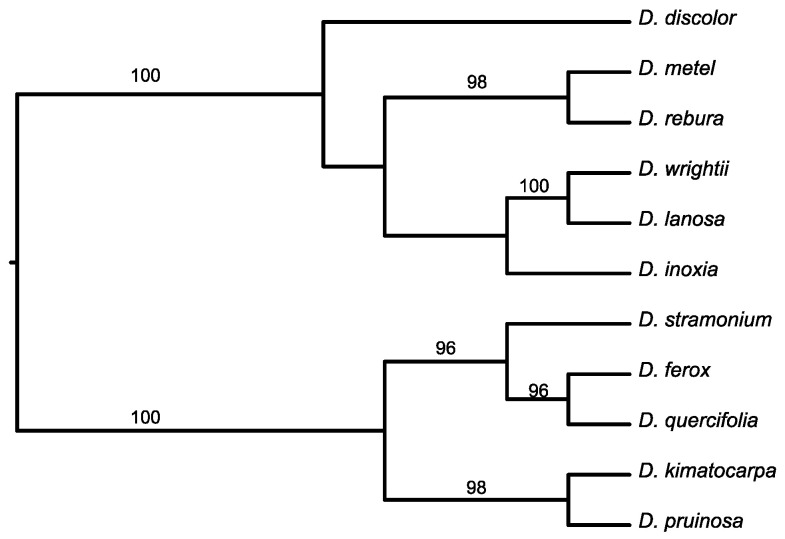
Ultrametric tree for the maximum likelihood phylogeny of the eleven species of *Datura* based on chloroplast (*psb*A-*trn*H and *trn*L-*trn*F regions) and nuclear (ITS rRNA) sequences. The numbers above branches indicate bootstrap values > 70 (Núñez-Farfán et al., in prep).

**Table 1 plants-12-02611-t001:** ANOVA of herbivore performance. Performance of the generalist herbivore *Spodoptera frugiperda* and the specialist herbivore *Lema trilineata daturaphila* was measured as food consumption (FC), number of damaged leaves (DL), biomass increment (BI), and growth efficiency (GE), across eleven *Datura* species. The plant (*Datura*) term is a random effect, whereas the herbivore (degree of specialization) is a fixed effect. A significant *Datura* × herbivore interaction indicates that herbivore performance varies as a function of its degree of specialization towards plant species. Significant values are indicated in bold (*p* < 0.05).

		FC		DL		BI		GE	
	**DF**	F	*p*	F	*p*	F	*p*	F	*p*
**Herbivore**	1	33.07	**<0.0001**	18.24	**<0.0001**	129.14	**<0.0001**	12.06	**0.0006**
***Datura* spp.**	10	6.45	**<0.0001**	9.01	**<0.0001**	7.16	**<0.0001**	1.03	0.416
***Datura* spp. × herbivore**	10	10.27	**<0.0001**	5.59	**<0.0001**	10.14	**<0.0001**	1.79	0.06
**Error**	416								
**Total**	437								

**Table 2 plants-12-02611-t002:** Factor loadings for phylogenetic principal components analysis (pPCA) of herbivore performance (FC: food consumption, BI: biomass increment, GE: growth efficiency, and DL: number of damaged leaves), plant nutrients (N: nitrogen, P: phosphorous, C: carbon, and W: water content), and plant defenses (trop: tropine, hyos: hoysciamine, scop: scopolamine, atrop: atropine, sola: solanine, totalk: total alkaloids, and trich: leaf trichomes). Numbers in bold indicate the weightier variables.

Plant Traits		
	PC1	PC2
**Defense**		
**trop**	0.465	−0.174
**hyos**	−0.048	**0.464**
**scop**	**0.908**	−0.228
**atrop**	**0.718**	0.259
**sola**	0.243	0.361
**totalk**	**0.943**	−0.038
**trich**	−0.188	0.154
**Nutrients**		
**N**	**−0.761**	0.076
**P**	0.071	−0.461
**C**	−0.302	0.414
**W**	0.686	**0.595**
**Herbivore Traits**		
**Specialist: *Lema trilineata daturaphila***		
**BI**	**−0.630**	**0.576**
**FC**	−0.216	0.381
**GE**	−0.023	−0.023
**DL**	**−0.670**	**0.485**
**Generalist: *Spodoptera frugiperda***		
**BI**	−0.284	**−0.803**
**FC**	−0.357	**−0.869**
**GE**	0.429	**−0.656**
**DL**	**−0.532**	**−0.785**

Larvae between instars 2–4 of the specialist herbivore *L. trilineata daturaphila* and generalist herbivore *S. frugiperda* were fed for 48 h with 11 *Datura* species in a controlled environment.

**Table 3 plants-12-02611-t003:** Correlation between components of herbivore performance (FC: food consumption, BI: biomass increment, DL: leaf damaged, and GE: growth efficiency) and plant traits (N: nitrogen, P: phosphorous, C: carbon, W: water, scop: scopolamine, totalk: total alkaloids, and trico: leaf trichomes) corrected by the phylogenetic relationships (PGLs) of *Datura*. For each response variable, Akaike´s information criterion (AIC) values and the P of the statistical model are shown. Same AIC value and P among variables indicates that they belong to the same model. For the full models of each performance trait see Appendix A.

Performance Trait (Response Variable)	Plant Trait (Predictor Variable)	Correlation Coefficient	AIC	*p* (Model)
	*Lema trilineata daturaphila*			
FC	P	−0.35 *	−19.9	0.041
FC	C	−0.013 *	−19.9	0.041
BI	scop	−0.252	−9.87	0.118
BI	trico	−0.079	−9.87	0.118
DL	scop	−0.166 *	−20.33	0.008
DL	trico	−0.074 *	−20.33	0.008
*Spodoptera frugiperda*				
BI	C	−0.251 *	−31.43	0.008
BI	W	−0.165 *	−31.43	0.008
GE	N	−0.284 *	−48.8	0.091
GE	W	−0.133 *	−48.8	0.091
GE	C	−0.87 *	−48.8	0.091
DL	P	0.625	−40.7	0.0009
DL	W	−0.92 *	−40.7	0.0009
GE	scop	0.02	−48.96	0.05
GE	trico	−0.274	−48.96	0.05
DL	totalk	−0.104 *	−31.31	0.038
DL	trico	−0.004 *	−31.31	0.038

* *p* < 0.05.

## Data Availability

GeneBank access numbers and tables with average values of variables are provided in the Appendix A.

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
