# Peer review of "Effect of Plant Defenses and Plant Nutrients on the Performance of Specialist and Generalist Herbivores of Datura: A Macroevolutionary Study"

_plants, 2023, doi:10.3390/plants12142611_

Round 1
Reviewer 1 Report
Firstly I commend the authors on a fine piece of work. The quality is high and it is clear a lot of careful work went into doing this study. The study is also very novel, detailed and valuable contribution to the field of plant insect interactions. The picture of the herbivores is also a great addition to the manuscript.
I have only two minor comments:
1) In Supplementary Figure 2 you display results for 33 tests between the specialist and generalist herbivore with the significance value set as P < 0.05. You should however correct for multiple testing by using a Bonferroni correction or equivalent which will result in significance threshold being set even lower. There are several more recent papers that suggest formulas that are not overly corrected. See:
Midway S, Robertson M, Flinn S, Kaller M. 2020. Comparing multiple comparisons: practical guidance for choosing the best multiple comparisons test. PeerJ 8:e10387 http://doi.org/10.7717/peerj.10387
2) There are two papers by Angela Moles in New Phytologist which deals with the subject in a global multi-data set analysis; I suggest you include at least one in the introduction as part of the contextualisation of the study.
Correlations between physical and chemical defences in plants: tradeoffs, syndromes, or just many different ways to skin a herbivorous cat? AT Moles, B Peco, IR Wallis, WJ Foley, AGB Poore, et al. - New Phytologist, 2013
Putting plant resistance traits on the map: a test of the idea that plants are better defended at lower latitudes. AT Moles, IR Wallis, WJ Foley, DI Warton, JC Stegen, et al. - New Phytologist, 2011
Author Response
Response to reviewers:
Reviewer 1
Reviewer’s comment 1 (C1): Firstly, I commend the authors on a fine piece of work. The quality is high and it is clear a lot of careful work went into doing this study. The study is also very novel, detailed and valuable contribution to the field of plant insect interactions. The picture of the herbivores is also a great addition to the manuscript.
Author’s response 1 (R1):
We appreciate your favorable opinion and comments to the manuscript.
C2. I have only two minor comments: 1) In Supplementary Figure 2 you display results for 33 tests between the specialist and generalist herbivore with the significance value set as P < 0.05. You should however correct for multiple testing by using a Bonferroni correction or equivalent which will result in significance threshold being set even lower. There are several more recent papers that suggest formulas that are not overly corrected. See:
Midway S, Robertson M, Flinn S, Kaller M. 2020. Comparing multiple comparisons: practical guidance for choosing the best multiple comparisons test. PeerJ 8:e10387 http://doi.org/10.7717/peerj.10387
R2:
Thanks for the comments and reference. Indeed, we performed planned comparisons between the means of Lema vs. Spodoptera performances. We used t-test and corrected the alpha by testing each null hypothesis at Alpha value/n (where n is the number of tests). We did this correction for each character (i.e., 11 tests). The results changed slightly, i.e., few significant tests resulted non-significant after the correction.
C3. 2) There are two papers by Angela Moles in New Phytologist which deals with the subject in a global multi-data set analysis; I suggest you include at least one in the introduction as part of the contextualization of the study.
Correlations between physical and chemical defences in plants: tradeoffs, syndromes, or just many different ways to skin a herbivorous cat? AT Moles, B Peco, IR Wallis, WJ Foley, AGB Poore, et al. - New Phytologist, 2013
Putting plant resistance traits on the map: a test of the idea that plants are better defended at lower latitudes. AT Moles, IR Wallis, WJ Foley, DI Warton, JC Stegen, et al. - New Phytologist, 2011
R3:
Thanks for the suggestion. We added Moles et al. 2013 to the Introduction and rewrote the paragraph to put into context covariation among multiple defense traits. The text now reads as follows: “The link between plant defense traits and herbivores´ performance [11–13] and insects’ adaptation to such traits has been experimentally demonstrated [10] but may be constrained by synergistic effects when defense is composed by multiple plant traits. Although evidence of synergisms and tradeoffs between physical and/or chemical defenses has been documented whether combinations of defence traits are adaptive seem to be context dependent (Moles et al. 2013). Also, whether single or a suite of defense plant traits (physical or chemical) act more effectively against specialist or generalist herbivores remains unclear [14]. The understanding of the mechanisms and processes underlying the evolution of plant-herbivore interaction and the evolution of diet breadth in insects makes necessary to assess the response of generalist and specialist herbivores to multiple plant traits, on specific plant lineages”.

Reviewer 2 Report
The reviewed manuscript deals with the actual problem of plant traits affecting herbivore fitness. The question is investigated in full-factorial experiment using two contrasting types of herbivores (specialist vs. generalist) and many species of Datura (Solanaceae). The results are corrected to phylogenetic relationships between tested plant species. The main finding is that performance of specialist herbivore depens more on defensive plant traits, whereas that of the generalist - on plant nutritive value.
Nevertheless, the used methods leave some doubts. However, my own experience in the studies of plant-herbivore interactions tells me that some of concerns and minor methodological flaws are inevitable in researches on such complex issues. So I believe that the results obtained by authors, are interesting for coevolutionary biologists, but their presentation should be improved.
Major comments.
1)The studied speciealist and generalist do differ drastically in many life history traits and phylogenetic positions, so we can not exclude that differences between Lema trilineata and Spodoptera frugiperda are because of some other factors except their diet range. This limitation should be clearly stated and discussed in the text.
2) The authors used only data about area of ingested food without any correction for possibly different specific leaf area (SLA). Without SLA, area is not a good of estimated ingested food mass.
3) The logic way from obtained results of PGSL (or pPCA too?) to the conclusion that specialist's performance depends on defensive traits, whereas generalist's performance - on nutritive quality, does not seem clear for me. I believe, that this conclusion is not so sound given different components of plant performance and presence of traits from the "opposite" set in the best fitted model.
3) "Colorimetric techniques" of element analysis must be sescribed in more details.
4) As far as I understand,the authors took data on defensive chemicals from literature. But these chemical contents can have great inter-species variation. Some data from used plants, or from the same population, would be better. If it is not possible to obtain (which can be the case), it should be clearly stated as limitation to this study.
5) How growth efficience can be negative if relative biomass increment is positive (e.g., the case of Datura inoxia in table S2)?
Minor comments.
1) Does Figure 1 contain results of one-way ANOVAs or of univariate results for two-way ANOVA? And what does whiskers in this plot mean?
2) What are numbers in Table 2? I suppose, they are factor loads, i.e. correlation coefficient? Besides, in the 1st pararaph of Section 2.1 the authros said that greater loads had "correlation coefficient > ±0.5" whereas they obviously mean absolute value of correlation coefficienr > 0.5, or "correlation coefficient > 0.5 OR < -0.5"
3) Some information about the experiment and studied variables shoud be given in the Introduction or in the beginning of the Results sections because now the reader can not understand Results at all before reading Methods section, which goes further.
4) It is stated in the main text that larvae were placed on adaxial side of leaves, whereas in the Supplementary materials it is stated that they are placed on abaxial side. Given the photo in Supplementary, the "abaxial" is a misprint.
The English is pretty good except some typos:
1) "plan defences" instead of "plant defence" in the 1st line in the 2nd paragraph in Introduction
2) Noctinidae instead of Noctuidae in the Section 4.1
3) Names of chemical elements sometimes begin from the capital.
Author Response
Response to reviewers:
Reviewer 2
C1. The reviewed manuscript deals with the actual problem of plant traits affecting herbivore fitness. The question is investigated in full-factorial experiment using two contrasting types of herbivores (specialist vs. generalist) and many species of Datura (Solanaceae). The results are corrected to phylogenetic relationships between tested plant species. The main finding is that performance of specialist herbivore depends more on defensive plant traits, whereas that of the generalist - on plant nutritive value.
Nevertheless, the used methods leave some doubts. However, my own experience in the studies of plant-herbivore interactions tells me that some of concerns and minor methodological flaws are inevitable in research on such complex issues. So I believe that the results obtained by authors, are interesting for coevolutionary biologists, but their presentation should be improved.
R1:
Thank you for the valuable comments on the manuscript and the constructive feedback. We have addressed the comments on the Methods and Discussion.
C2. Major comments.
The studied specialist and generalist do differ drastically in many life history traits and phylogenetic positions, so we cannot exclude that differences between Lema trilineata and Spodoptera frugiperda are because of some other factors except their diet range. This limitation should be clearly stated and discussed in the text.
R2:
We agree on the importance of discussing the strengths and limitations of this study. We have therefore added this information to the Methods (Bioassays) and Discussion section (first paragraph).
C3. The authors used only data about area of ingested food without any correction for possibly different specific leaf area (SLA). Without SLA, area is not a good of estimated ingested food mass.
R3:
Thanks for your comment. We agree that specific leaf area (leaf area/dry mass) is an important parameter of plant performance but, alas, we did not measure it on the experimental plants, and this is an aspect that warrants consideration in future studies in the genus. SLA is usually reported in plant physiology and studies of ecosystems productivity (Vile et al. 2006, Walker et al. 2014). In the context of plant defense SLA is associated to leaf mechanical defenses and to estimate leaf toughness (e. g., Agrawal and Fishbein, 2006, Kozlov et al. 2015, Pereira et al. 2020). Although leaf toughness has been linked to (poorer) herbivore performance (Clissold et al. 2009), the amount of consumed food (or leaf damage) has been used as a conventional estimate for insect performance and fitness, in empirical studies on plant defense in experimental and natural populations (e.g. Karban and Agrawal 2002, Johnson et al. 2009). This parameter has the advantage of capturing both the physiology and acceptance (behavior) of a specific food by an herbivore. In the present study, allows to confirm that besides host specificity, both herbivores consumed a substantial amount of tissue from different species of Datura, which validate the subsequent analysis.
C4. The logic way from obtained results of PGSL (or pPCA too?) to the conclusion that specialist's performance depends on defensive traits, whereas generalist's performance - on nutritive quality, does not seem clear for me. I believe, that this conclusion is not so sound given different components of plant performance and presence of traits from the "opposite" set in the best fitted model.
R4:
We do agree with your observation. Since different performance components of two herbivores were influenced by both plant defense- and nutrient-related traits, we rephrased the discussion on this matter and clarified whether a set of characters better predicted performance (based on score of permutation models) or weather several performance components involve combinations of most plant traits indicating a wide range of influence of such traits on herbivores´ performance.
For instance, models that included hyoscyamine and leaf trichomes, better predicted GE and FC of the specialist herbivore (Supplementary Table S3). In contrast, a wider set of plat nutritional traits were involved in significant correlations between performance traits of the generalist herbivore, including BI, GE and DL (Table 3).
Based on this notion, we rewrote the Discussion section (paragraphs #1,2) and the Abstract.
C5. "Colorimetric techniques" of element analysis must be described in more details.
R5:
We have added some specifics. The paragraph now reads as follows: “These samples were used to analyze Nitrogen (N), Phosphorus (P), and Carbon (C) using colorimetric techniques that included molybdenum blue colourimetric method and malachite green assay, preceded by the process of mineralization to obtain simple inorganic compounds (Szydłowska-Czerniak, and Szłyk, 2003)”.
C6. As far as I understand, the authors took data on defensive chemicals from literature. But these chemical contents can have great inter-species variation. Some data from used plants, or from the same population, would be better. If it is not possible to obtain (which can be the case), it should be clearly stated as limitation to this study.
R6:
We agree that chemical compounds have great intra- and inter-specific variation. In fact, data presented here, on plant nutrients and herbivores performance were obtained from the same plant populations/species studied for specialized metabolites described in Kariñho-Betancourt et al. (2015). We now have included this information in the Introduction section (last paragraph).
C7. How growth efficiency can be negative if relative biomass increment is positive (e.g., the case of Datura inoxia in table S2)?
R7:
A general result here presented, indicates that growth efficiency (GE) of the specialist herbivore (L. trilineata daturaphila) was positive based on (positive) estimation of biomass increment in all Datura species. Contrasting with this, the generalist herbivore showed negative biomass increment in seven out of eleven Datura species, including D. inoxia (Figure S2), resulting in an overall negative GE (Figure 1). The referred Table S2 displays the variance between herbivores within species, per each performance trait. We included the mean value of each herbivore, and for the generalist most mean BI values were negative, matching the displayed data of Figure S2.
C8. Minor comments.
Does Figure 1 contain results of one-way ANOVAs or of univariate results for two-way ANOVA? And what does whiskers in this plot mean?
R8:
We performed a t-student to compare the mean response between the two herbivores (for different variables). Whiskers depict the SEM (standard error of the mean)
C9. What are numbers in Table 2? I suppose, they are factor loads, i.e. correlation coefficient? Besides, in the 1st pararaph of Section 2.1 the authors said that greater loads had "correlation coefficient > ±0.5" whereas they obviously mean absolute value of correlation coefficienr > 0.5, or "correlation coefficient > 0.5 OR < -0.5"
R9: Yes, these values are the PCA loadings. We have pointed out this in the table title.
We rephrased this paragraph for clarity.
C10. Some information about the experiment and studied variables should be given in the Introduction or in the beginning of the Results sections because now the reader can not understand Results at all before reading Methods section, which goes further.
R10:
We agree, this is a real problem with the format of the Plants journal. We have now included specifics of the methods at the end of the Introduction and at the beginning of results (paragraphs # 1, 2 ,3 ,4) to provide some context.
C11. It is stated in the main text that larvae were placed on adaxial side of leaves, whereas in the Supplementary materials it is stated that they are placed on abaxial side. Given the photo in Supplementary, the "abaxial" is a misprint.
R11:
Thank you, we have corrected the typo in the Supplementary material.
Additional comment by the authors. Although not required by the reviewers, we provided the ultrametric version of the phylogeny as this is the conventional format in which this information is presented.
References
Agrawal, A. A., & Fishbein, M. (2006). Plant defense syndromes. Ecology, 87(sp7), S132-S149.
Clissold, F. J., Sanson, G. D., Read, J., & Simpson, S. J. (2009). Gross vs. net income: how plant toughness affects performance of an insect herbivore. Ecology, 90(12), 3393-3405.
Johnson, M. T., Smith, S. D., & Rausher, M. D. (2009). Plant sex and the evolution of plant defenses against herbivores. Proceedings of the National Academy of Sciences, 106(43), 18079-18084.
Karban, R., & Agrawal, A. A. (2002). Herbivore offense. Annual Review of Ecology and Systematics, 33(1), 641-664.
Kariñho‐Betancourt, E., Agrawal, A. A., Halitschke, R., & Núñez‐Farfán, J. (2015). Phylogenetic correlations among chemical and physical plant defenses change with ontogeny. New Phytologist, 206(2), 796-806.
Kozlov MV, Lanta V, Zverev V, et al. (2015) Background losses of woody plant foliage to insects show variable relationships with plant functional traits across the globe. Journal of Ecology 103:1519–28.
Pereira, C. C., Boaventura, M. G., de Castro, G. C., & Cornelissen, T. (2020). Are extrafloral nectaries efficient against herbivores? Herbivory and plant defenses in contrasting tropical species. Journal of Plant Ecology, 13(4), 423-430.
Vile, D., Garnier, E., Shipley, B., et al. (2005). Specific leaf area and dry matter content estimate thickness in laminar leaves. Annals of Botany, 96(6), 1129-1136.
Walker, A. P., Beckerman, A. P., Gu, Le., et al. (2014). The relationship of leaf photosynthetic traits–Vcmax and Jmax–to leaf nitrogen, leaf phosphorus, and specific leaf area: a meta‐analysis and modeling study. Ecology and Evolution, 4(16), 3218-3235.
Round 2
Reviewer 2 Report
The authors took into account all my comments, so now, on my mind, the manuscript is worth to be published, except two small typos I detected (see below).
I stiil believe that the authors generalize too much about specialist vs generalist performance, but their mention this limitation several times in the text, so readers could make their own oppinion.
I detected only two typos: "significative" instead of "significant" at the line 128 and "examin" instead of "examine" at the line 192
Author Response
Thanks for your help to improve the manuscript!